# MASTERING THE DUNGEON: GROUNDED LANGUAGE LEARNING BY MECHANICAL TURKER DESCENT

**Zhilin Yang, Saizheng Zhang, Jack Urbanek, Will Feng, Alexander H. Miller**
**Arthur Szlam, Douwe Kiela & Jason Weston**
Facebook AI Research

## ABSTRACT

Contrary to most natural language processing research, which makes use of static datasets, humans learn language interactively, grounded in an environment. In this work we propose an interactive learning procedure called Mechanical Turker Descent (MTD) and use it to train agents to execute natural language commands grounded in a fantasy text adventure game. In MTD, Turkers *compete* to train better agents in the short term, and *collaborate* by sharing their agents' skills in the long term. This results in a gamified, engaging experience for the Turkers and a better quality teaching signal for the agents compared to static datasets, as the Turkers naturally adapt the training data to the agent's abilities.

## 1 INTRODUCTION

Research in natural language processing often relies on static benchmark datasets, which are used to train models and measure the progress of the field. Human language, however, does not emerge from training on a language dataset, but from communication, and interaction with an environment. Interactive learning offers several advantages over static datasets, such as the ability for teachers and learners to control the data distribution according to the learner's abilities (Bengio et al., 2009), and the ability to pair the learning of language with the ability to act. That is, a language learning agent can learn to interact and communicate *with respect* to concepts that are grounded in a shared environment (Kiela et al., 2016).

In this work, we propose a general framework for interactive learning called *Mechanical Turker Descent* (MTD), which gamifies the collaborative training of machine learning agents over multiple rounds. MTD is a *competitive* and *collaborative* training protocol. It is competitive in that in each round, Turkers train their own agent to compete with other Turkers' agents to win the bonus. Due to the engaging nature of the competitive setting, Turkers are incentivized to create the best curriculum of training examples for their agents (not too easy, not too hard—but just right). At the same time, MTD is also collaborative, as Turkers' data are merged after each round and shared in the next round. As a result, the agents improve their language abilities by interaction with humans and the environment in the long term.

We demonstrate MTD in the setting of grounded language learning, where the goal is to teach agents to follow user directions in an interactive game interface called *GraphWorld*. The world is represented as a set of objects, along with directed typed edges indicating the relations between them. The set of possible actions of an agent are then defined as updates to the graph structure. While MTD is a general-purpose interactive learning procedure, it is particularly well-suited for this kind of scenario, where the grounded environment facilitates data efficiency and imposes constraints that make language learning easier and faster. It also allows for growing the complexity of the task as the learning agent improves.

Based on the GraphWorld interface, we build a text adventure game called *Mastering the Dungeon*, where humans train a dragon which lives in a dungeon and interacts with various objects (e.g. elven sword), containers (e.g. treasure chest), locations (e.g. tower), and non-player characters (e.g. trolls). Turkers give training example pairs $(x, y)$, where $x$ is a natural language command and $y$ is an action sequence. The task is formulated as a language grounding problem where agents are trained to learn the mapping from $x$ to $y$.

In our experiments, we set up variants of MTD along with a baseline of static data collection on Mechanical Turk. We show that for agents, either parameterized as standard Seq2Seq models with attention (Sutskever et al., 2014; Bahdanau et al., 2014), or as *Action-Centric Seq2Seq* (AC-Seq2Seq) models specially designed to take advantage of GraphWorld's structure, learning with MTD outperforms static training. Moreover, our ablation study shows that being engaging to humans and matching the training data distribution with the agent abilities are two important factors leading to the effectiveness of MTD.

## 2 RELATED WORK

Research into language learning can be divided into work that studies static datasets and work that studies grounding in an environment where learning agents can act. It is generally easier to collect natural language datasets for the former fixed case. Static datasets such as visual question answering (Antol et al., 2015) provide grounding into images, but no possibility for language learning through interaction. Some works utilize a geographical environment such as a maze but still employ static datasets (Artzi & Zettlemoyer, 2013).

It has been argued that virtual embodiment of agents is a viable long-term strategy for artificial intelligence research and the learning of natural language semantics, particularly in the form of games which also contain human players (Kiela et al., 2016). Grounding language in an interactive environment is an active area of research, however a number of recent works employ synthetic, templated language only (Sukhbaatar et al., 2015; Yu et al., 2017; Bordes et al., 2010; Hermann et al., 2017; Mikolov et al., 2015; Chaplot et al., 2017). Some works that do utilize real natural language and interaction include Wang et al. (2016), where language is learnt to solve block puzzles, and Wang et al. (2017) where language is learnt to draw voxel images, which are both quite different to our case of studying text adventure games. Other works study text adventure games, like we do, but without the communication element (He et al., 2016; Narasimhan et al., 2015).

Many methods that collect natural language for learning utilize Amazon's Mechanical Turk, as we do. However, the overwhelming majority collect data both in a static (rather than interactive) fashion, and by using the standard scheme of a fixed payment per training example; this includes those works mentioned previously. We use such collection schemes as our baseline to compare to the Mechanical Turker Descent (MTD) algorithm we introduce in this paper.

There are some systems that have attempted to apply competitive, collaborative and/or gamification strategies to collect data, notably the *ESP game* (Von Ahn & Dabbish, 2004), which is an image annotation tool where users are paired and have to "read each others mind" to agree on the contents of an image. *ReferItGame* (Kazemzadeh et al., 2014) and *Peekaboom* (Von Ahn et al., 2006) have similar ideas but for localizing objects. In a completely different field, *Foldit* is an online game where players compete to manipulate proteins (Eiben et al., 2012). In comparison, our approach, Mechanical Turker Descent, is not specific to a particular task and can be applied across a wide range of machine learning problems, whilst more directly optimizing the quality of data for learning.

## 3 ALGORITHM: MECHANICAL TURKER DESCENT

The Mechanical Turker Descent (MTD) algorithm is a general method for collecting training data. It is designed to be engaging for human labelers and to collect high quality training data, avoiding common pitfalls of other data collection schemes. We first describe it in the general case, and subsequently in Section 4.1 we describe how we apply it to our particular game engine scenario.

MTD consists of $N$ human labelers (Turkers) who all use a common interface for data collection, and a sequence of rounds of labeling, where feedback is given to the labelers after each round. Before the first round, we initialize two datasets $D_{train\_all}$ and $D_{test\_all}$, which could either be (i) empty; or (ii) initial sets of data collected outside of the MTD algorithm. Both $D_{train\_all}$ and $D_{test\_all}$ are shared by all the labelers and updated each round.

Each round consists of the following steps, also summarized in Figure 1:

1. At the beginnning of the round, each of the $N$ Turkers provides a set of labeled examples in the form of $(x, y)$ pairs, giving $N$ datasets $D_1, \ldots, D_N$. In our experiments we consider

Figure 1: The *competitive-collaborative* Mechanical Turker Descent (MTD) algorithm. In each round Turkers are *competitive* to produce the best training data. However, in subsequent rounds they share all the data from the previous rounds so they are *collaborative* in the long term. The shared datasets are omitted here for simplicity.

two settings for data collection: either (i) a fixed number of examples, or (ii) as many examples as the Turker can provide within a fixed time limit (with a lower bound on the number of examples). We find that (ii) is a more natural setup in order to avoid idle time due to stragglers, and to encourage individual engagement and efficiency.

2. In the next step, $N$ separate models are trained, one for each Turker, each using the same learning algorithm, but different data. For Turker $i$, a model $M_i$ is trained on the dataset $D_i \cup D_{train\_all}$.

3. Each Turker $i$ is assigned a score $S_i$ for the quality of their labeling based on the performance of their model $M_i$. The model $M_i$ is evaluated using accuracy (or some other evaluation metric) on the evaluation dataset $(D_1 \cup D_2 \cup \cdots \cup D_N \cup D_{test\_all}) \setminus D_i$, i.e. using the shared test set along with all other Turker's data other than their own. Let $|D_m| = \min_i |D_i|$ be the size of the smallest dataset. We propose to normalize the metric by the size of the datasets to avoid bias towards any one Turker's dataset. The score of Turker $i$ is computed as:

$$S_i = \frac{\sum_{j \neq i} |D_m| Acc(M_i, D_j) + |D_{test\_all}| Acc(M_i, D_{test\_all})}{(N-1) \cdot |D_m| + |D_{test\_all}|} \tag{1}$$

where $Acc(m, d)$ measures the accuracy for model $m$ on dataset $d$. The scores of the Turkers are made visible via a high-score table of performance. A paid bonus is awarded to the Turkers who have the top scoring entries in the table. This is an explicit gamification setting designed to engage and motivate Turkers to achieve higher scores and thus provide higher quality data.

4. The data from all Turkers collected in this round is added to the shared datasets. More specifically, we split the dataset $(D_1 \cup \cdots \cup D_N)$ randomly into two subsets $D_{train\_cur}$ and $D_{test\_cur}$, and update the shared datasets to make them available to all Turker's models on the subsequent round: $D_{train\_all} \leftarrow D_{train\_all} \cup D_{train\_cur}$ and $D_{test\_all} \leftarrow D_{test\_all} \cup D_{test\_cur}$. At this point, the process repeats.

MTD is a competitive-collaborative algorithm. In each round, Turkers are incentivized to provide better data than their competitors. However, evaluation of quality is measured by performance on datasets from competing Turkers, making it inherently collaborative: they must agree on a common "language" of examples, i.e. they must follow a similar distribution. Further, on subsequent rounds,

due to Step 4, they all share the same data they collected together, hence they are incentivized to work together in the longer term. After each round, the data of all $N$ Turkers are merged to train a single model: it is clear that having $N$ different models would make any of those models worse than a single model built collaboratively. Secondly, a Turker in the current round benefits from other Turkers in previous rounds, which ensures that worse-off Turkers from previous rounds can still compete. One can make an analogy with the publication model of the research community, where researchers competitively write papers to be accepted at conferences (which is like a round of MTD), whilst using each other's ideas to build subsequent research for the next conference (which is like subsequent rounds of MTD).

**Why is this a good idea?** Our algorithm simultaneously brings two advantages over standard data collection procedures. Firstly it gamifies the data collection process, which is known to be more engaging to labelers (Von Ahn & Dabbish, 2004). Secondly, our approach avoids many of the common pitfalls of conventional data collection, leading to high quality data:

- **Avoids examples being too easy** In standard data collection, there is nothing preventing new training examples being *too easy*. Many similar training examples may have already been collected, and a model may only need a subset of them to do well on the rest. In MTD, there is no incentive to add easy examples as these will not improve the Turker's trained model, which negatively affects their score (position in the leaderboard). In addition, since the data is also used to evaluate other Turkers' models, providing easy examples will lead to higher scores of other Turkers, yielding a competitive disadvantage.

- **Avoids examples being too hard** In standard data collection, there is nothing preventing new training examples being *too hard* for the model to generalize from. In MTD, there is no incentive to add too hard examples, as these will also not improve the Turker's model and their score.

- **Human-curated curriculum** In MTD, there is incentive to provide examples that are "just right" for the model to generalize well to new examples. As the model should be improving on each round, this also incentivizes Turkers to provide a curriculum (Bengio et al., 2009) of harder and harder examples that are suitable for the model as it improves. Since the Turker acts as the model's teacher they are essentially defining the curriculum as teachers do for students. Choosing the best examples for the model to see next is also related to active learning (Cohn et al., 1994) except in our case this is chosen on the teacher's, rather than the learner's side.

- **MTD is not easily exploitable/gameable** Mechanical Turk data collection is notorious for providing poor results unless the instructions and setup are very carefully crafted (Goodman et al., 2013). MTD's scoring system is resistant to a number of attacks designed to game it. Firstly, collusion is difficult as Turkers are randomly grouped into a set of $N$ participants with no ability to communicate or to find out who the other participants are. Even if collusion does occur between e.g. a pair of Turkers, as $N$ is expected to be large ($N = 30$ in our experiments) and the evaluation scores are averaged by Turker such an attack is of small influence (cf. Eq. 1). Secondly, if a Turker seeks to create an evaluation set that reduces other Turkers scores (by acting on their own or via collusion) e.g. by creating hard-to-classify examples, these examples are importantly also the ones their own model is trained on. Hence, this strategy is actually more likely to deteriorate their own model's performance, while having relatively small influence on the performance of others. In general optimal performance is found by cooperating with others to some degree (making examples somewhat similar) whilst being competitive (trying to make more and/or more useful examples than their competitors).

There are also a number of extensions one could consider to MTD, we describe some of them in Appendix I.

## 4 GAME ENVIRONMENT: MASTERING THE DUNGEON

In this section we describe a general game interface called *GraphWorld* that we employ in our experiments. It is designed to be modular and extensible. Being a game, it provides an engaging

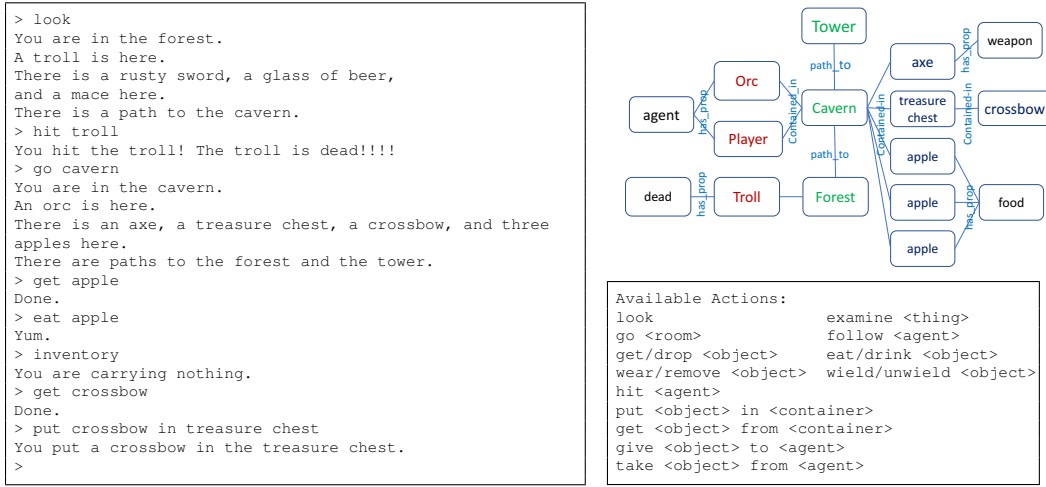

```
> look
You are in the forest.
A troll is here.
There is a rusty sword, a glass of beer,
and a mace here.
There is a path to the cavern.
> hit troll
You hit the troll! The troll is dead!!!!
> go cavern
You are in the cavern.
An orc is here.
There is an axe, a treasure chest, a crossbow, and three
apples here.
There are paths to the forest and the tower.
> get apple
Done.
> eat apple
Yum.
> inventory
You are carrying nothing.
> get crossbow
Done.
> put crossbow in treasure chest
You put a crossbow in the treasure chest.
>
```

```
Available Actions:
look                  examine <thing>
go <room>             follow <agent>
get/drop <object>     eat/drink <object>
wear/remove <object>  wield/unwield <object>
hit <agent>
put <object> in <container>
get <object> from <container>
give <object> to <agent>
take <object> from <agent>
```

Figure 2: Example gameplay from GraphWorld (left), part of the underlying graph representation (top right) and the set of actions possible within the game (bottom right).

interface between agents and humans for data collection, and focuses on research into grounded agents that learn to both communicate and act.

The underlying representation (grounding) in GraphWorld is a graph where each concept, object, location and actor is a node in the graph, and labeled edges represent relations between them. For example, paths between locations are edges with "path_to" labels, an agent is in the location which is connected to it with a "contained_by" edge, and movement involves altering the latter "contained_by" edge to another location. Similarly, objects have various properties: food, drink, wearable, wieldable, container, and so on. Each action in the game (if it can be executed, depending on the graph state) leads to a new state which is a change in the graph structure. Every action hence has a set of prequisites (e.g. is there a path in the graph that makes this move action possible) followed by a transformation of the graph that executes it.

Here, this underlying representation is used to generate a classic text adventure game, *Mastering the Dungeon*, in a fantasy setting with swords, castles and trolls, but it is a general formalism that could be used to build other games as well. The actions include moving, picking up objects, eating, etc. We implemented a total of 15 action types, which closely follow those of classical online text adventure MUD (multi-user dungeon) games such as DikuMUD[1]. The full list is given in Fig 2 along with an example of execution of the game and the underlying graph structure representation.

What is appealing about the GraphWorld formalism is that the grounding is extensible with (i) new actions, which can be coded by simply providing new transformations of the graph, and (ii) with more concepts – new locations, objects and actors can easily be added. This means that the (grounded) language in the simulation can easily grow, which is important for language research where small restricted dictionaries in simulations keep the research synthetic in nature (Weston et al., 2015). Here, we explore the mapping of natural language to grounded actions within GraphWorld, but the framework allows the study of other language and reasoning phenomena as well.

## 4.1 MTD for GraphWorld

We investigate the learning problem of mapping from a natural language command $x$ to a sequence of actions $y$ in GraphWorld, for example *"enter the bedchamber and toss your armor on the bed"* maps to *"go bedroom; remove helmet; put helmet on bed; remove chestplate; put chestplate on bed"*. We set up the MTD game as follows: each Turker is a player who is given a companion pet dragon that they can provide commands to. The player has to "train their dragon" by issuing it commands in natural language which it has to execute, and their goal is to train their dragon to perform better than their competitors, just as described in Section 3 in the general MTD case.

---

[1]https://en.wikipedia.org/wiki/DikuMUD

The particular interface for the Turkers we chose is the text adventure game itself, where they can type actions. To simplify the experience for novice gamers and first time users, at each step in the game, we list the set of possible actions so that the Turker can simply select one of them. At any stage (after any number of actions) they can enter "teach" to indicate that the last sequence entered will be the set of actions for a new training example (or else "reset" if they want to discard their current sequence). After entering "teach" the Turker provides the natural language command that should result in that set of actions. The natural language command and a representation of the state of the world become the input $x$ and the actions that should be executed become the output $y$ for the training example $(x, y)$. For representing the state of the world we simply store the entire graph, different models can then make use of that in different ways (e.g. represent it as features).

Data collection is performed within a randomized adventure game world (randomized for each training example) consisting of 3 locations, 3 agents, 14 objects (weapons, food, armor and others) and 2 containers, where locations and paths are randomized. We employ 30 Turkers on each round, and consider two settings: (i) ask them to create 10 examples each round; or (ii) ask them to create at least 10 examples each round (but they can create more) with a maximum time of 40 minutes. The length of the action sequence is constrained to be at most 4. For each example added, the existing trained model from the previous round is executed and the Turker is told if the model gets the example correct already (which implies that the example is possibly "too easy"). This can help the Turker enter useful examples for the subsequent model to train on. The leaderboard scores and bonus awarded (if any) are emailed at the end of each round. At that point, Turkers can sign up for the next round, which does not necessarily have to employ the same Turkers, but we did observe a significant amount of return players. We perform 5 rounds of MTD.

A natural comparison for MTD is the traditional method of data collection: simply pay Turkers per example collected. We choose the total pay to sum to the same as as the base pay plus bonuses for MTD, so the same dollar amount is spent. We ran this also as 5 rounds, but each round is effectively the same, as no model feedback is involved, and no leaderboard or bonuses are emailed. We also made sure that new Turkers were recruited, without prior game play experience of MTD, so as to avoid bias.

## 5  EXPLOITING GRAPHWORLD'S STRUCTURE: AC-SEQ2SEQ

Our agent aims to learn a mapping from natural language command $x$ to action sequence $y$. We treat this as a supervised learning problem. In the following text, we use "model" and "agent" interchangeably.

A natural baseline is the sequence-to-sequence (Seq2Seq) model (Sutskever et al., 2014) with attention (Bahdanau et al., 2014). That model, however, is generic and does not make full use of the characteristics of the task: taking into account some priors about our task may be more data efficient. With this in mind, we propose the *Action-Centric Seq2Seq* (AC-Seq2Seq) model, which takes advantage of the grounded nature of our task, specifically by incorporating inductive biases about the GraphWorld action space.

AC-Seq2Seq shares the same encoder architecture with Seq2Seq, in our case a bidirectional GRU (Chung et al., 2014). The encoder encodes a sequence of word embeddings into a sequence of hidden states. AC-Seq2Seq has the following additional properties: it models (i) the notion of actions with arguments (using an action-centric decoder), (ii) which arguments have been used in previous actions (by maintaining counts); and (iii) which actions are possible given the current world state (by constraining the set of possible actions in the decoder). Details are provided below.

**Compositional Action Representation**

Let $\mathcal{A}$ denote the action space. Each action in the action space $a \in \mathcal{A}$ can be denoted as $a = (type, arg_1, arg_2)$, which specifies a composition of an action type and two arguments. For example, the action *take elven sword from troll* is denoted as $(take\_from, elven\_sword, troll)$. For actions with one argument, $arg_2$ is set as none; i.e., *go tower* is denoted as $(go, tower, none)$.

AC-Seq2Seq utilizes a compositional representation for each action $a$. More specifically, we concatenate an action type embedding with two argument embeddings, i.e., $\mathbf{a} = [\mathbf{e}_{type}, \mathbf{e}_{arg_1}, \mathbf{e}_{arg_2}]$.

A compositional action representation is data-efficient because different actions share common action type and/or argument representations. For example, it is easier for the model to generalize to *get elven sword* after seeing *get treasure chest* because the *get* representations are shared. In contrast, the baseline Seq2Seq model treats each action in the action sequence as atomic, which neglects their compositional nature[2].

**Action-Centric Decoder**

First we describe how we vectorize the input before introducing the decoder formulations. Consider a decoding step $j$. For each action $a = (type, arg_1, arg_2)$, we employ the two argument embeddings $\mathbf{e}_{arg_1}$ and $\mathbf{e}_{arg_2}$ as query vectors to attend over encoder hidden states respectively, and concatenate the two attention results, denoted as $att_a$. Let $count_{a,j}$ be the number of occurrences of the two arguments $arg_1$ and $arg_2$ in previous decoding steps from 1 to $j - 1$. Let $location_j$ be the current location (e.g., cavern). We then use a graph context vector $env_{a,j}$ to encode $count_{a,j}$ and $location_j$ by concatenating their learnt embeddings. This idea is related to the checklist model of Kiddon et al. (2016).

A key difference between Seq2Seq and AC-Seq2Seq is that instead of using a *single* vector representation (hidden state) at each time step to predict an action, AC-Seq2Seq maintains a *set* of action-centric hidden states. More specifically, we maintain a hidden state $\mathbf{h}_{a,j}$ for action $a$ at decoding step $j$. The hidden states are updated as follows

$$\mathbf{h}_{a,j} = GRU([\mathbf{a}; \ att_a; \ env_{a,j}], \mathbf{h}_{a,j-1})$$

In other words, we concatenate an action representation $\mathbf{a}$, an attention result $att_a$, and a graph context vector $env_{a,j}$ as the input. A GRU is employed to update the hidden states for *each* action, and the weights of the GRU are shared among actions.

Given the model parameter $\mathbf{w}$, the probability distribution over the action space at decoding step $j$ can be written as

$$P_{a,j} = \frac{\exp \mathbf{w}^\top \mathbf{h}_{a,j}}{\sum_{a' \in \mathcal{A}} \exp \mathbf{w}^\top \mathbf{h}_{a',j}}$$

The above action-centric formulation allows us to leverage the compositionality of action representations described in Section 5. Moreover, such an action-centric view enables better matching between the input natural language commands and the action arguments because the attention mechanisms are conditioned on actions. For example, one can tie the embeddings of *tower* in the natural language command and *tower* in the action *go tower* so that it is possible for the model to learn *go tower* even without seeing the word *tower* before.

**Action Space Decoding Constraint** During decoding, we constrain the set of possible actions to be only among the valid actions given the current world state. For example, it is not valid to *go tower* if the dragon is in the tower, or there is no path to the tower from the current location. This constraint is applied to both Seq2Seq and AC-Seq2Seq in our experiments.

## 6 EXPERIMENTS

We employ the environment and MTD settings described in Section 4.1, code and data for which will be made available online.[3] For all the results in this section, we train the agents for 10 runs and report the mean and standard deviation. To study the effects of interactive learning, we compare the following learning procedures:

- **MTD** is our proposed algorithm. The Turkers are asked to create at least 10 examples per round (but they can create more) in a maximum time of 40 minutes, repeated for 5 rounds.
- **MTD ablations:** We consider two possible ablations of the MTD algorithm:
  - **MTD limit** has a limit on the number of examples. The Turkers are asked to create exactly 10 examples per round. Our hypothesis is that Turkers are willing to create

---

[2]Note, we also consider various ablations that test model variants that sit somewhere between our main Seq2Seq and AC-Seq2Seq models in the Appendix G.

[3]https://github.com/facebookresearch/ParlAI/tree/master/projects/mastering_the_dungeon

| Method | Accuracy | hits@1 | F1 |
|---|---|---|---|
| *Training AC-Seq2Seq* | | | |
| MTD | $0.418 \pm 0.010$ | $0.461 \pm 0.033$ | $0.701 \pm 0.009$ |
| MTD limit | $0.402 \pm 0.009$ | $0.431 \pm 0.033$ | $0.682 \pm 0.007$ |
| MTD limit w/o model | $0.386 \pm 0.010$ | $0.419 \pm 0.053$ | $0.682 \pm 0.007$ |
| *Collaborative-only* baseline | $0.334 \pm 0.015$ | $0.329 \pm 0.034$ | $0.644 \pm 0.012$ |
| *Training Seq2Seq* | | | |
| MTD | $0.261 \pm 0.005$ | $0.026 \pm 0.002$ | $0.589 \pm 0.008$ |
| MTD limit | $0.241 \pm 0.003$ | $0.024 \pm 0.003$ | $0.569 \pm 0.006$ |
| MTD limit w/o model | $0.229 \pm 0.003$ | $0.020 \pm 0.002$ | $0.554 \pm 0.005$ |
| *Collaborative-only* baseline | $0.219 \pm 0.005$ | $0.032 \pm 0.003$ | $0.525 \pm 0.010$ |

Table 1: Main evaluation results. Interactive learning (MTD) outperforms static learning (collaborative-only baseline).

more examples to win the game (be higher on the leaderboard), hence MTD limit should be worse than MTD.

– **MTD limit w/o model** is *MTD limit* without model feedback. The Turkers are not informed about the model predictions and thus cannot adapt the data distribution according to the agent abilities. Our hypothesis is thus that MTD and MTD limit should outperform this method.

• **Collaborative-only baseline** is the conventional static data collection method where Turkers are asked to create 10 examples given a fixed amount of payment. Total payment is set to be equal to the MTD variants.

During online deployment of the MTD algorithm, AC-Seq2Seq models are trained each round and deployed to inform the Turkers about model predictions, to evaluate the agents' performance and to produce the leaderboard ranking. We combine the shared test sets $D_{test\_all}$ from all of the above settings and an initial pilot study dataset (see Appendix E) to form a held-out test set for all methods. We train Seq2Seq models using the same training data collected using AC-Seq2Seq models[4], and evaluate them on the same held-out test set.

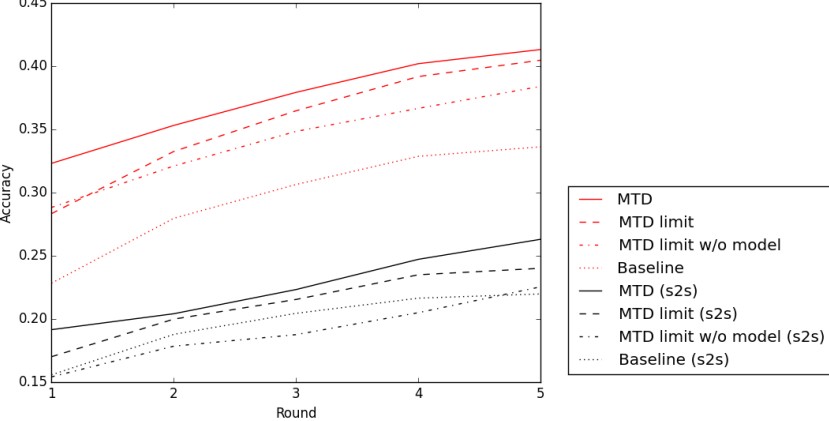

Figure 3: Learning curves of different methods. Red lines and black lines correspond to AC-Seq2Seq and Seq2Seq respectively.

We report three metrics: accuracy, hits@1 and F1. Accuracy is determined by the ratio of test examples for which the action sequence predicted by the model leads to a correct end state as defined by the underlying graph. To compute hits@1 for each test example $x$, we randomly sample 99

---

[4]This gives a fair comparison for the collaborative-only baselines between the two models, but an unfair advantage to AC-Seq2Seq for MTD, however we were limited in the number of Turk jobs we could run.

additional examples in the test set and compute the rank of $y$ from within that list; hits@k for larger $k$ are given in the Appendix, Fig. 3. F1 is defined at the action level and averaged over examples. The results are given in Table 1.

MTD outperforms static data collection (collaborative-only baseline) substantially and consistently on all the metrics for both models. The improvement over the collaborative baseline is up to 8.4 points in accuracy and 13.2 points in hits@1. This indicates that MTD is effective at collecting high-quality data and thus training better agents. Unsolicited feedback from Turkers also indicates their high level of engagement, see the Appendix H for details.

The ablation study shows that MTD outperforms MTD limit, which shows that through an engaging, gamified setting, Turkers have higher incentives to create more examples in order to win the competition, and create 30% more examples on average compared to MTD limit. Both MTD and MTD limit outperform MTD limit w/o model. This clearly indicates that model feedback contributes to better agent performance. This also justifies our argument that dynamic coordination between training data distribution and agent abilities is important, avoiding too easy or too hard examples.

Lastly, AC-Seq2Seq outperforms Seq2Seq by a large margin of up to 15.7 points in accuracy, demonstrating that the inductive biases based on the GraphWorld action space are important. Similar trends can also be observed in Fig 3, where we plot the learning curves of agents trained with different learning procedures. We examine the relative contribution of (i) tracking which arguments have been used in previous actions and (ii) which actions are possible given the current world state (by constraining the set of possible actions in the decoder) in a separate ablation study in Appendix G, and find that these lead to improved performance.

## 7 CONCLUSIONS

We studied the interactive learning of situated language, specifically training agents to act within a text adventure game environment given natural language commands from humans. To train such agents, we proposed a general interactive learning framework called Mechanical Turker Descent (MTD) where Turkers train agents both collaboratively and competitively. Experiments show that (i) interactive learning based on MTD is more effective than learning with static datasets; (ii) there are two important factors for its effectiveness: it is engaging to Turkers, and it produces training data distributions that match agent's capabilities. Going forward, we hope to apply these same techniques to learn more complex language tasks in richer domains.

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

## A  EXAMPLES OF COLLECTED DATA

| Natural language command $x$ | Action sequence $y$ |
|---|---|
| steal the crown from the troll and put it on | take silver crown from troll wear silver crown |
| pick up the leather pouch and put the armor in it | get leather pouch put armor in leather pouch |
| walk to the forest and hand the blue ring to the troll | go forest give blue ring to troll |
| drink some beer, and then walk through the forest to the tower | drink beer go forest go tower |
| pick up the armor and put it on | get armor wear armor |
| wear the gold ring and put the beer inside the treasure chest before you go to the cavern | get gold ring wear gold ring put beer in treasure chest go cavern |
| equip the rusty sword. go to the cavern and kill the orc with the rusty sword | get rusty sword wield rusty sword go cavern hit orc |
| eat one of the apple and give the other apple to the orc | get apple get apple give apple to orc eat apple |
| kill the troll with the axe before putting axe in treasure chest | get axe hit troll put axe in treasure chest |
| take the blue and gold ring and head to the forest to give the troll the gold ring | get gold ring get blue ring go forest give gold ring to troll |
| fly into the cavern. find an apple and eat it and then strip the gold ring from the orc | go cavern get apple eat apple take gold ring from orc |
| feed the troll with your bread and beer. fly up to the tower and find your axe within the magical treasure chest. | give bread to troll give beer to troll go tower get axe from treasure chest |
| dress yourself with the gold ring and acquire the silver crown. place it upon your head and then ready your rusty sword for battle. | wear gold ring get silver crown wear silver crown wield rusty sword |
| disarm the troll then hit it. | take rusty sword from troll hit troll |
| grab the crossbow and race to the cavern. steal the beer from the troll and chug it | get crossbow go cavern take beer from troll drink beer |

Table 2: Examples of collected data.

Examples of collected data are shown in Table 2.

## B  MTURK INSTRUCTIONS AND INTERFACE

The MTurk instructions and interface are shown in Figs 4 and 5 respectively.

## C  RANKING EVALUATION METRICS

In addition to Table 1, hits@k for $k = 5$ and $k = 10$ are reported in Table 3. With AC-Seq2Seq, MTD consistently outperforms the baseline, and also the ablation study results are consistent to Table 1. Seq2Seq is shown to be ineffective at ranking action sequences and gives poor performance on both MTD and the baseline.

## D  ACCURACY BREAKDOWN BY OUTPUT LENGTH

We report accuracy breakdown by output length (number of actions in the sequence $y$ for a given input $x$) in Table 4. Agents trained on the collaborative-only baseline data are slightly more effective at learning length 1 examples, but fail to generalize to longer action sequences. For example, Seq2Seq trained by MTD outperforms Seq2Seq trained on the baseline data by up to 180% on length

| Method | hits@5 | hits@10 |
|---|---|---|
| *Training AC-Seq2Seq* | | |
| MTD | $0.733 \pm 0.023$ | $0.788 \pm 0.019$ |
| MTD limit | $0.720 \pm 0.022$ | $0.779 \pm 0.018$ |
| MTD limit w/o model | $0.715 \pm 0.030$ | $0.778 \pm 0.026$ |
| *Collaborative-only* baseline | $0.655 \pm 0.020$ | $0.740 \pm 0.013$ |
| *Training Seq2Seq* | | |
| MTD | $0.107 \pm 0.005$ | $0.152 \pm 0.005$ |
| MTD limit | $0.098 \pm 0.006$ | $0.150 \pm 0.004$ |
| MTD limit w/o model | $0.096 \pm 0.005$ | $0.145 \pm 0.004$ |
| *Collaborative-only* baseline | $0.123 \pm 0.006$ | $0.165 \pm 0.004$ |

Table 3: Ranking evaluation metrics.

| Method | Length 1 | Length 2 | Length 3 | Length 4 |
|---|---|---|---|---|
| *Training AC-Seq2Seq* | | | | |
| MTD | 0.638 | 0.465 | 0.329 | 0.226 |
| MTD limit | 0.603 | 0.440 | 0.322 | 0.232 |
| MTD limit w/o model | 0.601 | 0.396 | 0.300 | 0.229 |
| *Collaborative-only* baseline | 0.655 | 0.383 | 0.182 | 0.094 |
| *Training Seq2Seq* | | | | |
| MTD | 0.568 | 0.243 | 0.122 | 0.071 |
| MTD limit | 0.544 | 0.231 | 0.099 | 0.054 |
| MTD limit w/o model | 0.504 | 0.185 | 0.105 | 0.079 |
| *Collaborative-only* baseline | 0.600 | 0.146 | 0.043 | 0.025 |

Table 4: Accuracy breakdown by output length (number of actions in sequence $y$ for a given input $x$).

4 examples. Note that MTD workers are trying to optimize their score which is an average over the entire distribution, so will not be directly aware of or trying to optimize for this breakdown.

## E    PILOT STUDY

In order to develop and improve our MTurk instructions and our model, we conducted a pilot study. We randomly generated sequence actions based on a uniform distribution, and asked the Turkers to use natural language to describe the sequence actions. We collected 400 samples in total. This initial pilot study dataset is also randomly split into two subsets and used to initialize $D_{train\_all}$ and $D_{test\_all}$ respectively in MTD.

## F    ACCURACY BREAKDOWN BY DATASETS

In Table 1, we report the results on a combined test set. Here we also report the accuracy on different test sets obtained by different training procedures, including the pilot study dataset, the baseline test

| Method | Test on MTD limit | Test on Baseline | Test on Pilot Study |
|---|---|---|---|
| *Training AC-Seq2Seq* | | | |
| MTD limit | 0.418 | 0.478 | 0.377 |
| *Collaborative-only* baseline | 0.343 | 0.461 | 0.348 |
| *Training Seq2Seq* | | | |
| MTD limit | 0.264 | 0.355 | 0.222 |
| *Collaborative-only* baseline | 0.237 | 0.347 | 0.219 |

Table 5: Accuracy breakdown on datasets. The first column indicates how the model is trained, and the first row indicates which test set the model is evaluated on.

set, and the MTD test set. The results are given in Table 5. It is clear that MTD consistently outperforms the baseline on all test sets. The margin on the MTD test set is larger than on the baseline test set, because the baseline test set has a more similar distribution to the baseline training set (which the baseline agent is trained on).

## G MODEL ABLATIONS

| Model | Accuracy | F1 |
|---|---|---|
| AC-Seq2Seq | 0.418 | 0.701 |
| AC-Seq2Seq w/o counter | 0.367 | 0.668 |
| AC-Seq2Seq w/o counter w/o location | 0.382 | 0.686 |
| Seq2Seq | 0.261 | 0.589 |

Table 6: Ablation study on model variants.

We remove the counter feature $count_{a,j}$ and the location embedding $location_j$ subsequently from AC-Seq2Seq and evaluate the performance. The results are reported in Table 6. Maintaining a counter of previous arguments contributes substantially to our final performance, which indicates the importance of a compositional action representation. However, encoding the location information is not beneficial, which suggests that the agent has not yet learned to utilize such information. Lastly, even without the counter, AC-Seq2Seq still substantially outperforms Seq2Seq, demonstrating the effectiveness of our action-centric decoder architecture.

| Method | Acc w/ DC | Acc w/o DC | F1 w/ DC | F1 w/o DC |
|---|---|---|---|---|
| *Training AC-Seq2Seq* | | | | |
| MTD | $0.418 \pm 0.010$ | $0.271 \pm 0.020$ | $0.701 \pm 0.009$ | $0.574 \pm 0.024$ |
| MTD limit | $0.402 \pm 0.009$ | $0.254 \pm 0.020$ | $0.682 \pm 0.007$ | $0.554 \pm 0.022$ |
| MTD limit w/o model | $0.386 \pm 0.010$ | $0.242 \pm 0.016$ | $0.682 \pm 0.007$ | $0.551 \pm 0.019$ |
| *Collaborative-only* baseline | $0.334 \pm 0.015$ | $0.203 \pm 0.016$ | $0.644 \pm 0.012$ | $0.489 \pm 0.019$ |
| *Training Seq2Seq* | | | | |
| MTD | $0.261 \pm 0.005$ | $0.173 \pm 0.004$ | $0.589 \pm 0.008$ | $0.472 \pm 0.010$ |
| MTD limit | $0.241 \pm 0.003$ | $0.152 \pm 0.005$ | $0.569 \pm 0.006$ | $0.445 \pm 0.010$ |
| MTD limit w/o model | $0.229 \pm 0.003$ | $0.154 \pm 0.006$ | $0.554 \pm 0.005$ | $0.449 \pm 0.009$ |
| *Collaborative-only* baseline | $0.219 \pm 0.005$ | $0.156 \pm 0.007$ | $0.525 \pm 0.010$ | $0.399 \pm 0.022$ |

Table 7: Ablation study: model performance with (w/) and without (w/o) action space decoding constraint. "Acc" stands for *accuracy*, and "DC" means *decoding constraint*.

To further study the effects of utilizing the action space on model performance, we compare the performance with and without the action space constraint during the decoding phase. Results are reported in Table 7. It is clear that action space constraints improve the performance substantially for all settings with both models.

## H FEEDBACK FROM TURKERS

We observed positive, unsolicited feedback from Turkers, including the following quotes, for instance on Turkerhub[5]:

- "I actually got one of the Train Your Dragon hits, and it was glorious! That is, a whole lot of fun, as I started to figure it out. Whatever the case, really, really awesome task. Was so cool."
- "Interesting and even, dare I say, fun."
- "Having accepted it, I read through the long instructions on the left and thought, "Wow! This looks awesome!" even though it's a mite confusing and certainly a lot to absorb at once."

---

[5]https://turkerhub.com

- "You teach a dragon to do stuff like an old text adventure game."
- "They are kind of fun. Took me awhile to understand what I needed to do."
- "Can I do another hit like this?"
- "Just want to see what the hype is about with them. Also, who doesn't like dragons?"
- "They are pretty fun."
- "The non-competitive ones are fairly easy and quick; just did my first two today as well. But want that bonus real bad."
- "I wonder if he finally realized that we have zero reason to put any effort into the non bonus versions of his hit though. My work for those was low effort compared to my min maxing on the bonus ones."

This supports the hypothesis that MTD is an engaging gamified experience for humans. Particularly the last two quotes tend to support that the competitive gamification is far more engaging (note that this is before we banned Turkers from doing both tasks as we wanted to avoid bias, and hence had to redo all the experiments).

## I    EXTENSIONS TO MTD

One can consider a number of modifications and extensions to the MTD approach, we list a few important ones here.

- **Within-round model feedback** Turkers receive indirect feedback about the quality and abilities of the model they are training via the MTD scores they receive each round. However, more explicit feedback can also be given. We consider in our experiments to run every new example $(x, y)$ through the existing model (trained from the previous round) and to inform the Turker about the prediction of that model. This will give valuable online feedback *within round* on how the Turker should shape their dataset. For example, if the example is already correctly classified, this is a warning not to create examples like this. Additionally, one could provide the Turker with examples of performance of the model on data from the previous round, either from the Turker themselves or from others. We decided against the latter in our experiments as it introduced complexity and requires more skill on the part of the Turkers to understand and use the information, but we believe with experienced engaged human labelers, such information could be valuable.

- **Removing poor quality data** To deal with the problem of a given Turker $i$ entering very poor quality data, one can introduce automatic approaches to cleaning the data. This could be important so that these examples are not added in Step 4 to $D_{train\_all}$ or $D_{test\_all}$[6]. Clearly poor data will be reflected in a low score $S_i$ which has already been computed, so if this is very low relative to other models then it can be directly used as a filter. We suggest to compare it to models trained from the previous round, and to remove if it is inferior.

- **Dealing with high-dimensional input spaces** In a very rich learning problem, there may be an issue that Turkers label very different parts of the input space, leading to all Turkers obtaining low scores. One solution would be on each round to suggest a "subject area" (part of the input space) for all Turkers to focus on for that round.

- **From round-based to fully online MTD** As the shared dataset $D_{train\_all}$ is always increasing each round, online incremental learning initialized from the model trained from the previous round could be used, which is an active area of research (Rusu et al., 2016). However, for simplicity and transparency, in our experiments we trained from scratch (on the entire dataset up to that point) at each iteration of MTD. Fully online learning also gives the intriguing possibility of removing the notion of rounds, and making the scoring fully online as well, which could possibly be a more engaging experience.

---

[6]However, in our experiments, none of the data was this low quality

## Basic Setup and Payment

You will play a fantasy adventure game where you have to teach your dragon (a computer bot) human language!

The competition has multiple rounds. Each HIT is a round where 30+ Turkers compete against each other.

You have to provide **10 valid examples** to receive the base payment. **You will not get paid if any of your examples is invalid or if you do not finish 10 examples in 40 minutes**. No auto-rejection! The HIT will get expired if you don't type anything for 8 mins.

After each round, we will have a ranking of Turkers based on how well their dragons understand human language. High-ranked Turkers will receive bonuses in addition to the base payment:
rank 1 --- $13
rank 2 - 4 --- $8
rank 5 - 8 --- $3
rank 9 - 19 --- $1

## Actions, Orders, and Examples

An **action** is what the dragon can do. An **order** is an instruction in human language (English).

An **example** consists of an **order** and the corresponding **actions**. For instance, examples can look like:
**Actions**: get axe     **Order**: collect the axe
**Actions**: go cavern     **Order**: fly to the cavern
**Actions**: hit orc     **Order**: kill the orc
**Actions**: get axe go tower put axe in treasure chest     **Order**: pick up the axe, climb up the tower, and hide the axe in the treasure chest

## How to Teach

You will teach the dragon with the following input formats:
You: first_action
You: second_action
...
You: last_action
You: **teach**
You: order

For instance, you can enter the following inputs:
You: go cavern
You: **teach**
You: run to the cavern

You: eat apple
You: go cavern
You: **teach**
You: eat the apple and fly to the cavern

You have to enter **teach** 10 times to give 10 examples (in order to get paid).

## Your Strategy for Winning the Competition

Don't provide **too simple** or **too hard** examples! Good examples are those **slightly harder than what your dragon can do right now**.

### Strategy #1
The **more actions** an example has, the **harder** it is; e.g., eat apple go cavern is harder than go cavern.

### Strategy #2
Actions involving **two** objects are **harder**; e.g. give axe to troll is harder than drop axe.

### Strategy #3
Usually your dragon will get higher scores if you **increase the variety** of your orders; e.g., using **"pick up"**, **"collect"**, and other different orders for the same action **get**.

If you are ready, please click "Accept HIT" to start this task.

Figure 4: MTurk instructions.

**You**: drop axe

**Dungeon Master**:
Done.

----------------------------------------------------

**Actions so far:**

drop axe

**What to enter now?**

You can enter **one** of the following:

1. You can enter **teach** now to finish the actions. You have to enter **teach** 10 more times to get paid. Actions of any lengths are valid. Go ahead and teach!

2. You can enter **exactly one** of the following actions (just copy and paste!):
wear gold ring, give gold ring to orc, go forest, get axe, get apple, get armor, hit orc, drop gold ring

3. You can enter **reset** to discard the previous actions and start from a new one.

**You**: teach

**Dungeon Master**: Your actions drop axe resulted in the following changes:

axe -> cavern

**Write an order in one or two sentences to precisely describe the above changes.** You can enter **reset** to discard the above changes and start from new actions. Try increasing the variety of your language, and **DO NOT** directly copy the actions!

Figure 5: MTurk interface. "Dungeon master" is a computer program that acts as the interface of our text adventure game, and "You" refers to the Turker.

