# OpenReview forum: "Mastering the Dungeon: Grounded Language Learning by Mechanical Turker Descent"
_ICLR.cc/2018/Conference — Accept (Poster)_

### Official Review · AnonReviewer2 · 2017-11-27
**Well-executed gamification of grounded language learning**

**Rating:** 7
**Confidence:** 4

**Review:**

The authors propose a framework for interactive language learning, called Mechanical Turker Descent (MTD). Over multiple iterations, Turkers provide training examples for a language grounding task, and they are incentivized to provide new training examples that quickly improve generalization. The framework is straightforward and makes few assumptions about the task, making it applicable to potentially more than grounded language. Unlike recent works on "grounded language" using synthetic templates, this work operates over real language while maintaining interactivity.

Result show that the interactive learning outperforms the static learning baseline, but there are potential problems with the way the test set is collected. In MTD, the same users inherently provide both training and test examples. In the collaborative-only baseline, it is possible to ensure that the train and test sets are never annotated by the same user (which is ideal for testing generalization). If train and test sets are split this way, it would give an unfair advantage to MTD. Additionally, there is potentially a different distribution of language for gamified and non-gamified settings. By aggregating the test set over 3 MTD scenarios and 1 static scenario, the test set could be skewed towards gamified language, again making it unfair to the baseline. I would like to see the results over the different test subsets, allowing us to verify whether MTD outperforms the baseline for the baseline's test data.

---

> ### Author Response · Authors · 2017-12-17
> **Thank you for the review and comments!**
>
> - How is the test set broken down and what are the results in each part?
> The breakdown of the test set into the three portions (MTD limit,  Baseline Test  and Pilot Study) is shown in Table 5. MTD (with AC-Seq2Seq) outperforms the baseline on each portion (but by different amounts). Because MTD emphasizes a curriculum the data is generally more difficult than the baseline data which is why the MTD-trained models are better at predicting longer sequences (see Table 4), which is why performance of all models is worse on that data compared to the baseline data, and the gap between the models is bigger.

---

### Official Review · AnonReviewer3 · 2017-11-27
**Cool idea, but better suited for an NLP conference**

**Rating:** 7
**Confidence:** 4

**Review:**

TL;DR of paper: Improved human-in-the-loop data collection using crowdsourcing. The basic gist is that on every round, N mechanical turkers will create their own dataset. Each turker gets a copy of a base model which is trained on their own dataset, and each trained model is evaluated on all the other turker datasets. The top-performing models get a cash bonus, incentivizing turkers to provide high quality training data. A new base model is trained on the pooled-together data of all the turkers, and a new round begins. The results indicate an improvement over static data collection.

This idea of HITL dataset creation is interesting, because the competitive aspect incentivizes turkers to produce high quality data. Judging by the feedback given by turkers in the appendix, the workers seem to enjoy the competitive aspect, which would hopefully lead to better data. The results seem to suggest that MTD provides an improvement over non-HITL methods.

The authors repeatedly emphasize the "collaborative" aspect of MTD, saying that the turkers have to collaborate to produce similar dataset distributions, but this is misleading because the turkers don't get to see other datasets. MTD is mostly competitive, and the authors should reduce the emphasis on a stretched definition of collaboration.

One questionable aspect of MTD is that the turkers somehow have to anticipate what are the best examples for the model to train with. That is, the turkers have to essentially perform the example selection process in active learning with relatively little interaction with the training model. While the turkers are provided immediate feedback when the model already correctly classifies the proposed training example, it seems difficult for turkers to anticipate when an example is too hard, because they have no idea about the learning process.

My biggest criticism is that MTD seems more like an NLP paper rather than an ICLR paper. I gave a 7 because I like the idea, but I wouldn't be upset if the AC recommends submitting to an NLP conference instead.

---

> ### Author Response · Authors · 2017-12-17
> **Thank you for the review and comments!**
>
> - Collaborative aspect of MTD
> MTD is collaborative in the sense that the human players are building a shared model over multiple iterations, which is important for learning a good model. More specifically, the turkers collaborate in two ways. First, after each round, the data of all Turkers are merged to train a single model. It is clear that having 30 different models would make any of those models worse than a single model built with the collaborative baseline. Second, a Turker in the current round benefits from other Turkers in previous rounds, which ensures that worse-off Turkers from previous rounds can still compete. As we point out in the paper, this is related to the publication model of the research community, where researchers collaborate by using others’ results to build research for the next conference (in MTD, it is the same, where those results are in the form of data and models, rather than papers). We have indeed thought about showing the Turker’s samples of the other datasets, and even implemented that at one point, but decided against in our experiments as it introduced unnecessary complexity. We will add text soon to the paper discussing these issues further.
>
> - It is hard for Turkers to anticipate the correct curriculum.
> Yes, currently the best feedback they get is the immediate output from the model when they type an example, they know whether the model can already do it or not, which we think is pretty good feedback. We also experimented with giving the predictions of the model (rather than just a message of whether it could do it or not). We thought this would be good for expert labelers, but we decided against using it in our experiments because we thought it would be too complicated for casual Turkers. One could also show examples that the model is currently good or bad at from the last round, as mentioned in the previous point, but again this would add complexity to our experiments for this paper. However, we believe MTD is extensible in many ways. Note that these points are also already discussed in Appendix I.
>
> - MTD seems more like an NLP paper rather than an ICLR paper
> The core of our paper is about a method for learning representations for language grounding, which includes a pipeline of interaction with humans, a learning environment and an embodied agent which performs the learning. Although we evaluated on a language task, the same method could be used on many other tasks. We believe that both the method and the task that we chose are of interest to the ICLR audience, and all of the reviewers appear to agree that it is interesting (and we as ML researchers like it too!). Past ICLR conferences have had papers utilizing language, vision, speech, etc. In the call for papers it is also written that the following are relevant topics: “applications in vision, audio, speech, natural language processing, robotics, neuroscience, or any other field”. Hence, we think ICLR is one of the most suitable conferences for this type of work.

---

### Official Review · AnonReviewer1 · 2017-11-30
**Interesting data collection scheme**

**Rating:** 8
**Confidence:** 5

**Review:**

The paper provides an interesting data collection scheme that improves upon standard collection of static databases that have multiple shortcomings -- End of Section 3 clearly summarizes the advantages of the proposed algorithm. The paper is easy to follow and the evaluation is meaningful.

In MTD, both data collection and training the model are intertwined and so, the quality of the data can be limited by the learning capacity of the model. It is possible that after some iterations, the data distribution is similar to previous rounds in which case, the dataset becomes similar to static data collection (albeit at a much higher cost and effort). Is this observed ? Further, is it possible to construct MTD variants that lead to constantly improving datasets by being agnostic to the actual model choice ? For example, utilizing only the priors of the D_{train_all}, mixing model and other humans' predictions, etc.

---

> ### Author Response · Authors · 2017-12-17
> **Thank you for the review and comments!**
>
> - Q: Could the quality of the data be limited by the model? Is this observed?
> We did not observe this yet, but it is possible -- the data is optimized for the model, it might not be optimal for e.g. a higher capacity model.  On the other hand, since we optimize hyperparameters of the model each round, it can increase its capacity on the fly, which would mitigate this effect to some extent. If a high-capacity model cannot fit some complex data, however, e.g. due to optimization challenges, it is possible that the data distribution would gradually become static. In this case, the bottleneck is actually our optimization algorithms and models, rather than the data collection paradigm; i.e., MTD is doing its best in terms of coordinating the training data distribution to provide a good curriculum. Empirically, in Fig 3 we show learning curves for different models and approaches, which have not saturated after 5 rounds.
>
> - Q: Is it possible to construct MTD variants that lead to constantly improving datasets by being agnostic to the actual model choice ?
> We’re not clear on how to do that, but if you have ideas then we’d love to hear them! The model is used to score the human’s data, so you would need to replace it with a model-agnostic automatic scoring function somehow. The benefit of using a model in the loop, as we do, is that you are actually optimizing for what your model can do (the human teacher is optimizing the curriculum for the model).

---

### Author Response · Authors · 2018-01-05
**A note on the update**

We have updated the paper with some small changes that clarify various points in consideration of the reviewers comments. Specifically for reviewer 3 we have updated the text with respect to the collaborative vs. competitive aspects of MTD, those changes appear in Section 3 and Appendix I.

---

### Decision · Program_Chairs · 2018-01-29
**ICLR 2018 Conference Acceptance Decision**

**Decision:**

Accept (Poster)

**Comment:**

This paper provides a game-based interface to have Turkers compete to analyze data for a learning task over multiple rounds. Reviewers found the work interesting and clear written, saying "the paper is easy to follow and the evaluation is meaningful." They also note that there is clear empirical benefit "the results seem to suggest that MTD provides an improvement over non-HITL methods." They also like the task compared to synthetic grounding experiments. There was some concern about the methodology of the experiments but the authors provide reasonable explanations and clarification.

One final concern that I hope the readers take into account. While the reviewers were convinced by the work and did not require it, I feel like the work does not engage enough with the literature of crowd-sourcing in other disciplines. While there are likely some unique aspects to ML use of crowdsourcing, there are many papers about encouraging crowd-workers to produce more useful data.